# Multi-LLM and Multi-Prompt Strategies for COVID-19 Infodemic Detection in Chinese Social Media: An Empirical Evaluation

**AI scientists**
AI Research Agent
Large Language Model

**Teng Zuo**
School of Health Policy and Management
Chinese Academy of Medical Sciences & Peking Union Medical College
Beijing, China
zuoteng@student.pumc.edu.cn

**Hongwen Lin**
School of Software Engineering
Xiamen University of Technology
Xiamen, China
2412114440@stu.xmut.edu.cn

**Lingfeng He**
Institute for Empirical Social Science Research
Xi'an Jiaotong University
Xi'an, China
helingfeng@stu.xjtu.edu.cn

**Hongji Zeng**
School of Health Policy and Management
Chinese Academy of Medical Sciences & Peking Union Medical College
Beijing, China
zhengzhouzhj@qq.com

**Lina Tang**
School of Social Research
Renmin University of China
Beijing, China
leana@ruc.edu.cn

**Li He**
Institute for Empirical Social Science Research
Xi'an Jiaotong University
Xi'an, China
lihegeo@xjtu.edu.cn

**Ning Li**
Department of Urology
Fourth Affiliated Hospital of China Medical University
Shenyang, China
ningli@cmu.edu.cn

## Abstract

**Objective:** Misinformation during the COVID-19 infodemic poses a serious public health risk. We investigate whether large language models (LLMs) can automatically identify COVID-19 misinformation in Chinese social media content, and how different prompting strategies affect performance. **Methods:** We evaluate ten LLMs on 640 physician-verified misinformation posts from a prior mixed-methods study (March 2022-October 2023). Each model issues a five-level predicted verdict (False / Likely-False / Ambiguous / Likely-True / True) under five prompting strategies (no-role; public-health expert; respiratory specialist; public-health ex-

pert + source/date context; respiratory specialist + source/date context). A single Qwen judge (`qwen-turbo-latest`) maps model responses to one of the five labels. We report strict accuracy (credit only False), lenient accuracy (credit False or Likely-False), ambiguity rate (Ambiguous), error rate (Likely-True / True), and a composite score. **Results:** Across all experiments, the average lenient accuracy was 61.2%, with a low overall ambiguity rate (<2%). Performance was highly model-dependent: the top-performing configuration achieved approximately 90% lenient accuracy, while more conservative models incorrectly accepted over 50% of false posts. Counterintuitively, prompting with expert personas and contextual details did not uniformly improve performance and, in many cases, reduced the models' flagging rates. **Contributions:** (1) An empirical, multi-LLM, multi-prompt evaluation on a previously established Chinese COVID-19 misinformation corpus. (2) A systematic comparison of five prompt strategies, quantifying how adding source/date context tends to reduce flagging on this all-misinformation benchmark while modestly lowering ambiguity. (3) Evidence that persona choice (public-health vs respiratory specialist) is not uniformly beneficial across posts and prompts. (4) A reproducible release (prompts, code, judging templates, redacted logs) to support Chinese-language infodemic monitoring and future replication.

# 1 Introduction

The COVID-19 "infodemic"—a rapid spread of false or misleading content—undermined public-health efforts [1, 2]. Manual monitoring by experts is too slow, narrow in coverage, and costly to scale across platforms, topics, and languages [3]. Large language models (LLMs) offer a plausible alternative: they can screen content at scale and in multiple languages. Yet LLMs are prompt-sensitive, may hallucinate [4], and can inherit training biases, motivating a systematic evaluation of accuracy, consistency, and prompt design.

Most prior work targets English or focuses on QA/advice generation rather than misinformation detection in Chinese social media [5]. We address this gap by repurposing a physician-verified Weibo corpus of 640 posts labeled as misinformation (Mar 2022-Oct 2023) as an evaluation bedrock (no new data collected). For each post, multiple state-of-the-art LLMs are prompted to issue a five-level veracity judgment—False, Likely-False, Ambiguous, Likely-True, True. A single Qwen judge (`qwen-turbo-latest`) maps outputs to these labels to enable consistent scoring.

We organize the study around four questions:

**Q1 (Feasibility):** Do off-the-shelf LLMs exceed a uniform-guessing baseline (20% over five options) on Chinese COVID-19 misinformation?

**Q2 (Model Differences):** How do models compare on strict/lenient accuracy, ambiguity, a composite score, and error (false acceptance)?

**Q3 (Prompting & Interaction):** Do expert-persona and source/date-context prompts improve accuracy or reduce ambiguity, and do effects depend on topic (public-health vs. respiratory medicine)?

**Q4 (External Robustness):** Does performance vary with posting time?

We test two hypotheses: H1—an "expert + detail" prompt (role plus source/date) increases strict accuracy and lowers ambiguity versus a zero-prompt baseline; H2—a respiratory-specialist persona particularly benefits respiratory-themed content. To probe these, we evaluate five prompting regimes: no role; public-health expert; respiratory specialist; and each expert persona augmented with source/date context.

# 2 Methodology

## 2.1 Dataset: Chinese Social Media Misinformation Dataset (CSMID)

We use the Chinese Social Media Misinformation Dataset (CSMID), established in the prior work [6]. The Chinese Social Media Misinformation Dataset (CSMID) comprises 236,775 public Weibo posts

in total. From this corpus, we extracted all physician-verified misinformation posts collected between March 2022 and October 2023, resulting in 640 items in total. Thus, the ground truth is binary and positive for every item ("misinformation"). We do not add new data, relabel, or reproduce raw posts. Topic tags and inclusion criteria follow the prior study; readers are referred to its appendix for full details.

**Ethics and governance.** Ethics approval was obtained from the Ethics Committee of the Fourth Affiliated Hospital of China Medical University (EC-2024-KS-211). The original CSMID was created under the IRB approval from the prior study.

## 2.2 LLM Model Pool and Settings

We evaluate 10 LLMs spanning proprietary and open(-ish) families and both English-centric and Chinese models, all zero-shot with no task-specific tuning. Runtime IDs (late-2024-early-2025 endpoints): GPT-4o (`gpt-4o`), Gemini-2.5-Pro (`gemini-2.5-pro-preview-06-05`), Mistral-Large (`mistral-large-latest`), Llama-4 "Maverick" (`meta-llama/llama-4-maverick`), Qwen3-235B (`qwen3-235b-a22b`), Qwen3-235B (thinking) (`qwen3-235b-a22b-thinking-2507`), DeepSeek-V3 (`deepseek-v3`), DeepSeek-R1 (`deepseek-r1-250528`), GLM-4 AirX (`glm-4-airx`), Doubao "Seed" (`doubao-seed-1-6-250615`).

All generations were produced between late-2024 and early-2025. Default decoding: temperature = 0.5, `max_tokens` $\approx 8000$. Prompts and outputs are in Chinese; provider safety settings are defaults.

## 2.3 Experimental Design: Prompting and Judging

Our overall workflow is as follows: for each of the 640 misinformation posts, we query each of the 10 LLMs using 5 distinct prompt strategies. This results in 32,000 unique responses. Each response is then evaluated by a single, fixed "judge" LLM, which assigns it one of the five veracity labels.

**Prompt Engineering Strategies** The manner in which a query is posed to an LLM (the prompt) can significantly alter the model's output. Prompt engineering techniques have been developed to guide LLM reasoning and reduce errors [7]. Role-playing prompts ask the model to adopt a persona or perspective (e.g., "You are a public health expert") to encourage domain-appropriate responses. Recent work found that such role prompts can influence accuracy: one study showed ChatGPT's COVID-19 misinformation detection accuracy dropped when multiple social identities were injected into the prompt, highlighting how certain roles or biases can degrade performance [13]. Conversely, a well-chosen role might improve focus—e.g., an infectious disease expert persona could make the model more critical of unverified claims. We test this by assigning expert personas to the models.

Another technique is inserting contextual details into the prompt to anchor the model's responses. By providing the social media post's source (e.g., "a Weibo user") and date, we aim to remind the LLM to consider the temporal context (important for COVID-19 facts that evolved over time) and the informal source, which might warrant skepticism. We test this by augmenting expert personas with contextual details to examine their joint effect.

Based on these principles, each model processes every post under five fixed prompts (identical across models):

**S1:** No-role baseline ("analyze for errors/misinformation").

**S2:** Public-health expert persona.

**S3:** Respiratory-specialist persona.

**S4:** Public-health expert + source/date context (Weibo user + post date).

**S5:** Respiratory specialist + source/date context.

For S4-S5 we use post dates from CSMID (temporal reporting uses quarter bins). Prompts ask for a structured verdict plus brief rationale.

**Automated Evaluation Pipeline (LLM-as-Judge)** Evaluating open-ended outputs of LLMs is challenging; thus, using an LLM to judge another LLM's output has emerged as a practical solution [8]. Recent research suggests that advanced models like GPT-4 can approximate human evaluation

of language tasks with high agreement (80%+ alignment with human judges in some settings). Such LLM-based evaluators (also called "LLM-as-a-judge") have been applied to compare chatbot answers and even to grade factual consistency in summaries [9]. In our study, we employed a single commercial Qwen judge (qwen-turbo-latest) as the evaluator, masking candidate identities to reduce bias and calibrating against a human gold standard. The judge prompt was provided in Chinese, and decoding parameters were fixed to ensure determinism (temperature = 0.1, max_tokens = 10, which are built-in control settings of the model). Nonetheless, we interpret the LLM-judge results with caution, as certain subtle errors may mislead the judge. The judge then maps each candidate response to one of five labels reflecting its stance, as shown in (Table 1).

Table 1: Predicted veracity labels used by the judge (applied to candidate LLM responses).

| Code | Judge label | Definition (what the response conveys) |
|------|-------------|----------------------------------------|
| 1 | False | Asserts the claim is incorrect or harmful with definitive language. |
| 2 | Likely-False | Indicates the claim is probably incorrect, expressing uncertainty or citing insufficient evidence. |
| 3 | Ambiguous | Declines to render a clear verdict, provides contradictory statements, or requests more information. |
| 4 | Likely-True | Indicates the claim is probably correct, but with qualifications or caveats. |
| 5 | True | Asserts the claim is correct/accurate with definitive language and alignment with consensus or guidelines. |

## 2.4 Evaluation Metrics

Because all CSMID items involved in this study are physician-verified misinformation, correctness is with respect to judge labels {1,2}. Let $y_j \in \{1, \ldots, 5\}$ be the judge label for post $j$ and $N = 640$ per condition:

- **Strict Accuracy** $\text{ACC}_{\text{strict}} = \frac{1}{N} \sum \mathbf{1}[y_j = 1]$.
- **Lenient Accuracy** $\text{ACC}_{\text{lenient}} = \frac{1}{N} \sum \mathbf{1}[y_j \in \{1, 2\}]$.
- **Error Rate** $\text{Err} = \frac{1}{N} \sum \mathbf{1}[y_j \in \{4, 5\}]$ (false acceptance).
- **Ambiguity** $\text{Amb} = \frac{1}{N} \sum \mathbf{1}[y_j = 3]$.
- **Composite Score** $\frac{2N_1 + 1N_2 - 1N_4 - 2N_5}{N} \in [-2, 2]$, rewarding decisive correct negatives and penalizing decisive mistakes (Ambiguous contributes 0).

## 2.5 Implementation and Reproducibility

The pipeline is implemented in Python; all inference is conducted via managed APIs (no local GPU). We fix decoding hyperparameters and freeze prompts. Vendor drift may affect exact regeneration, so we release cached generations and judgments, judge and candidate prompts, batching and rate-limiting scripts, and redacted logs (including runtime IDs, timestamps, and decoding parameters). For throughput, we use a client-side rate limit (500 req/min, token bucket), `ThreadPoolExecutor` (`max_workers=3`), and up to 3 retries with exponential backoff.

**Data governance:** we reuse the physician-verified CSMID corpus under the prior IRB; raw Weibo text is not redistributed (...examples deferred to the prior study's appendix). Statistical analysis and plots use standard Python stacks (pandas/numpy/statsmodels/matplotlib); Exact scripts and aggregated outputs are included for transparency.

# 3 Results

## 3.1 Aggregate Behavior on the CSMID Corpus

Figure 1 summarizes aggregate performance across the 50 model-prompt conditions (N = 32,000). Averaged across all models and prompts, the strict accuracy reached 48.0% and lenient accuracy was 61.2%. The ambiguity rate was consistently low, under 2% overall. This indicates that, on average,

LLMs can feasibly detect misinformation well above chance levels (answering Q1 positively), but with significant variation that warrants deeper analysis.

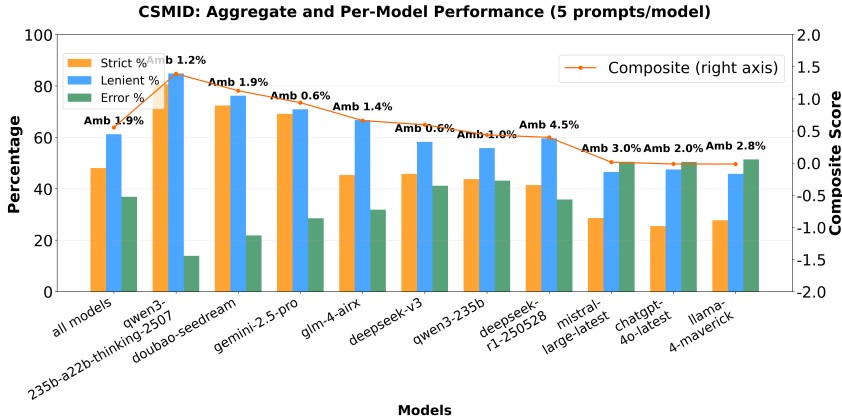

Figure 1: Aggregate performance across all 50 model-prompt conditions.

## 3.2 Variation Across Models (Averaged Over Prompts)

Per-model trends, as visualized in the heatmap of Figure 2, reveal distinct behaviors (addressing Q2). Assertive models like Qwen-thinking, Doubao-Seed, and Gemini-2.5-Pro consistently achieve higher composite scores (e.g., Qwen-thinking at 1.575 under S1) by flagging more decisively as False or Likely-False. In contrast, conservative models such as GPT-4o, Llama-4 Maverick, and Mistral-Large show lower scores (e.g., GPT-4o at -0.244 under S5), often accepting misinformation with labels 4 or 5. Ambiguity remains low across the board ($\sim$1-3%), with DeepSeek-V3 exhibiting the lowest rates (<1% in several regimes). These differences highlight model-specific biases: Chinese-centric models (e.g., Qwen, Doubao) tend to outperform English-centric ones on this Chinese corpus, possibly due to better linguistic and cultural alignment.

## 3.3 Prompting effects within models

To address Q3, we examine prompting effects in detail. Table 2 compares S1 (no-role) vs. S5 (respiratory-specialist + source/date) within each model, reporting percentage-point changes in lenient accuracy and ambiguity. For most models, adding expert personas and context reduces flagging rates (e.g., GLM-4 AirX drops from 1.283 to 0.188 in composite score), counter to H1. This suggests that contextual cues may encourage models to hedge or accept claims by considering temporal evolution or source informality. However, ambiguity often decreases modestly (e.g., GPT-4o from 4.1% to 2.0%), indicating improved decisiveness at the cost of accuracy on this benchmark.

Box plots in Figure 3 further illustrate cross-model dispersion per strategy. Expert-only prompts (S2, S3) yield higher medians for lenient accuracy (around 65-70%) compared to context-augmented ones (S4, S5; around 55-60%). This pattern holds across metrics, with error rates rising by 5-10 percentage points when context is added, particularly for models like Gemini (-13.8 pp in strict accuracy).

## 3.4 Persona Choice vs. Domain (S3 vs. S2)

Comparing respiratory-specialist (S3) vs. public-health expert (S2) personas, lenient accuracy differences are mostly negative (e.g., GPT-4o -10.0 pp, Gemini -8.1 pp), with small positives for Llama-4 (+1.1 pp) and Mistral-Large (+0.9 pp). This rejects H2, as the respiratory persona does not consistently benefit respiratory-themed content; instead, it may introduce domain-specific conservatism.

## 3.5 Temporal Stability

Addressing Q4, Figure 4 plots strict accuracy by anonymized quarter (2022 Q1–2023 Q4). Performance shows an early peak in 2022 Q2 ($\approx$ 75–82%), followed by a sharp decline in Q3 and

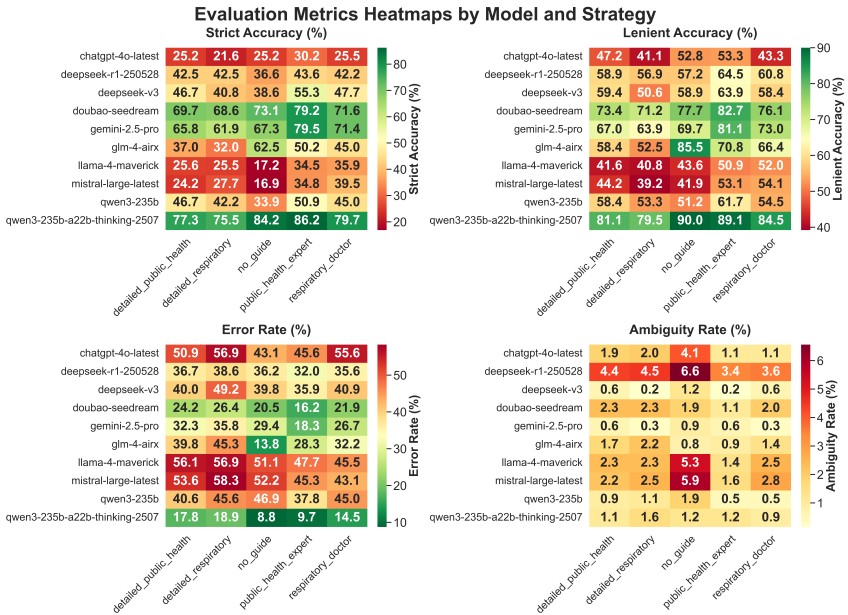

Figure 2: Evaluation metrics heatmaps by model and strategy. Four panels show Strict Accuracy, Lenient Accuracy, Error Rate (labels 4/5), and Ambiguity Rate (label 3). Rows are models; columns are the five prompting strategies (no-role, public-health expert, respiratory specialist, public-health+time/id, respiratory+time/id). Brighter cells in the strict/lenient panels and darker cells in the error panel indicate stronger flagging.

Table 2: Performance changes (composite score) and ambiguity rates from S1 to S5.

| Model | Score Change | Ambiguity |
|---|---|---|
| ChatGPT-4o | (0.133 → -0.244) | 4.1% → 2.0% |
| DeepSeek-R1 | (0.331 → 0.311) | 6.6% → 4.5% |
| DeepSeek-V3 | (0.550 → 0.373) | 1.2% → 0.2% |
| Doubao-Seed | (1.152 → 0.961) | 1.9% → 2.3% |
| Gemini-2.5-Pro | (0.898 → 0.637) | 0.9% → 0.3% |
| GLM-4 AirX | (1.283 → 0.188) | 0.8% → 2.2% |
| Llama-4 Maverick | (-0.108 → -0.203) | 5.3% → 2.3% |
| Mistral-Large | (-0.148 → -0.197) | 5.9% → 2.5% |
| Qwen3-235B | (0.241 → 0.348) | 1.9% → 1.1% |
| Qwen3-235B-Thinking | (1.575 → 1.195) | 1.2% → 1.6% |

stabilization around 40–55% through 2023 Q1–Q3, with a modest rebound in Q4 ($\approx$ 50–70%). These trends indicate relative temporal stability after the initial drop, with no evidence of systematic drift across quarters. The results suggest that models remain broadly robust to evolving COVID-19 knowledge during this period, though longer-term studies are required for confirmation.

### 3.6 Ambiguity and Error Tendencies

Ambiguity is extremely low (median 1.5%, see Figure 3), with the lowest rates for DeepSeek-V3 and Gemini-2.5-Pro ($\leq$1% across multiple regimes, as shown in the ambiguity panel of Figure 2). Error rates (false acceptances, labels 4/5) are higher for conservative models like GPT-4o, Llama-4 Maverick, and Mistral-Large ($\approx$50%, see error rate panel in Figure 2), while assertive models such as Doubao-Seed, Gemini-2.5-Pro, GLM-4 AirX, and Qwen3-235B-Thinking exhibit lower error rates (14-32%, Figure 3). These tendencies, visualized in the error rate distributions of Figure 3, underscore the trade-off between precision and recall in deployment, where conservative models

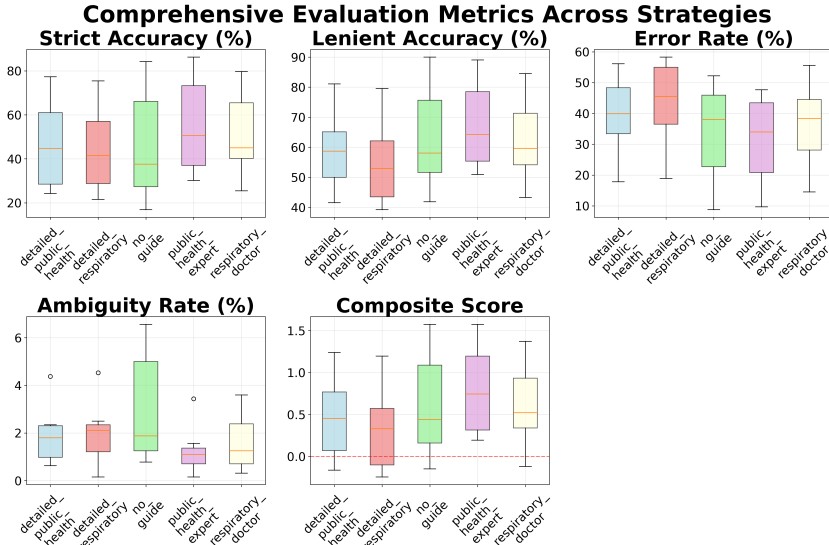

Figure 3: Comprehensive evaluation metrics across strategies (box plots). For each strategy, box plots summarize cross-model distributions of Strict, Lenient, Error, Ambiguity, and Composite Score. Medians and IQRs show that expert-only prompts (no time/id) tend to yield higher strict/lenient and lower error, while adding time/id nudges models toward acceptance (higher error, similar or slightly lower ambiguity). The composite-score panel includes a zero reference for interpretability.

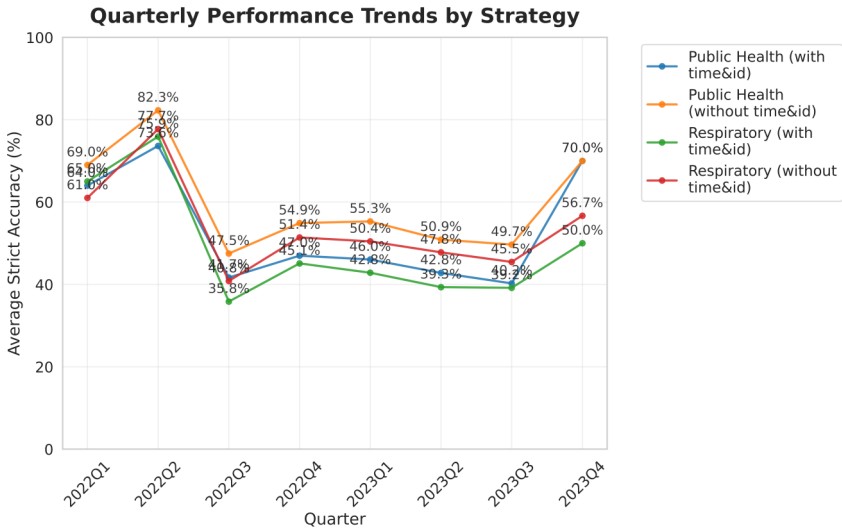

Figure 4: Quarterly trends in strict accuracy by model and strategy (`strategy_quarterly_trends`).

prioritize avoiding false positives at the cost of missing misinformation, while assertive models flag more aggressively with lower false acceptance rates.

## 4 Discussion

Health-related misinformation on social platforms has been studied extensively since the pandemic, with prior work exploring supervised classifiers, crowd-sourcing pipelines, and real-time "infoveillance" that monitors streams like Twitter or Weibo for emerging rumors. While rule-based and network-oriented signals (e.g., rumor cascades, provenance trails) can help [10], these systems

typically require labeled exemplars or substantial human oversight and tend to struggle when claims evolve. Recent surveys argue for interdisciplinary approaches that combine NLP with epidemiology and network analysis, yet the scale and velocity of online diffusion—often amplified by recommender systems and echo chambers [11]—still outpace manual or rigid rules-based detection. Against this backdrop, large language models (LLMs) offer an alternative or complementary route because they can, in principle, infer the veracity of previously unseen claims by drawing on broad background knowledge. LLMs have already shown promise across medical NLP tasks—achieving strong scores on medical exams, powering clinical QA, and enabling domain-tuned systems such as Med-PaLM [12]—while simultaneously raising concerns about reliability, hallucinations, and recency gaps.

Our study contributes to this literature by evaluating LLMs on detecting false versus true medical claims in Chinese social media posts, a setting that stresses not only factual recall but also robustness to layperson phrasing, incomplete context, and potentially deceptive content. Aggregated across models and prompts, strict and lenient accuracies reach 48.0% and 61.2%, respectively—well above uniform-guess baselines (20% and 40%). The best single configuration in our runs—the "thinking" Qwen3-235B variant with no-role prompting—achieves $\approx$84% strict and $\approx$90% lenient, suggesting that contemporary models can deliver decisive, largely correct negative judgments on an all-misinformation corpus with low ambiguity. In contrast, several frontier models (e.g., GPT-4o, Llama-4, Mistral-Large) accept false claims more often (labels 4/5 $\approx$50%), reflecting a more conservative stance.

Prompting matters, but not uniformly in the desired direction. We hypothesized (H1) that adding source/date context and expert personas would improve strict accuracy; in practice, these additions typically reduced flagging (hurting lenient accuracy on this benchmark) and only modestly decreased ambiguity, with Qwen3-235B as a notable exception. Swapping domain personas (respiratory specialist vs. public-health expert) did not yield consistent gains—only two models improved by $\approx$1 point—underscoring that prompt design should be tightly coupled to application objectives: when aggressive flagging is required, extra contextualizing cues may inadvertently nudge some models toward acceptance.

Failure modes and limitations remain. First, our results rely on an LLM-as-judge protocol; although the judge is fixed and tuned for determinism, it may encode systematic preferences. Second, ground truth in our evaluation is binary and uniformly positive (all posts are physician-verified misinformation), emphasizing flagging vs. acceptance rather than a symmetric five-way calibration. Third, the study focuses on one language (Chinese) and one domain/time window (COVID-19, Mar 2022-Oct 2023), so generalization is unknown. Finally, provider APIs evolve; we mitigate via cached generations, fixed prompts/decoding, and released logs, but perfect replay is not guaranteed.

Practical implications follow. For monitoring pipelines, more assertive models (e.g., Qwen-thinking, Doubao, Gemini, GLM-4 AirX) can yield higher flag rates with minimal ambiguity, but they should be paired with human-in-the-loop review to control over-flagging. Conversely, conservative models may be preferable where minimizing false positives is paramount. A pragmatic operational compromise is a multi-model ensemble—e.g., majority vote with a calibrated tie-break—which can balance precision and recall under shifting rumor distributions.

Future work should replace the single-judge with multi-judge consensus (with periodic human adjudication), incorporate retrieval-augmented checking to address recency and provenance, expand beyond COVID-19 and into multilingual settings, and—subject to privacy constraints—add coarse temporal/spatial stratification once anonymized metadata are releasable.

## 5 Conclusion

We present a multi-model, multi-prompt evaluation of LLMs for detecting COVID-19 misinformation in Chinese social media, using a previously established, physician-verified corpus. Models achieve low ambiguity and, in the best configuration, $\approx 84\%$ strict and $\approx 90\%$ lenient accuracy on this all-misinformation benchmark. Prompting with expert personas and context does not universally improve outcomes and can reduce flagging; task-aware prompt design is therefore essential. We release prompts, code, and redacted logs to support replication and further study; raw posts remain governed by the prior study's protocol. Overall, LLMs are not a substitute for human fact-checking, but they are a practical force multiplier for infodemic monitoring.

## AI Agent Setup

Our AI-assisted workflow leveraged a multi-agent, multi-LLM strategy for manuscript preparation. The initial draft was generated using an agent built on ChatGPT, referred to as `deep research`. All Python code for the experimental pipeline and data analysis was generated by Anthropic's Claude 4 Sonnet. The Qwen model family was employed to assist with data analysis and to brainstorm interpretations of the results. Grok was utilized to verify literature citations, demonstrating high accuracy in this role. Finally, Google's Gemini 2.5 Pro was used for iterative refinement and polishing, enhancing the manuscript's overall clarity, tone, and narrative flow.

## Responsible AI & Ethics Statement

This research was conducted with an emphasis on responsible AI use and complies with the NeurIPS Code of Ethics. We obtained ethics/IRB approval for using the CSMID dataset (which consists of public social media content) and adhered to the dataset license and platform Terms of Service; no new human subjects or paid crowd work were recruited. The primary aim of our work is to counteract harmful misinformation; however, we acknowledge the ethical challenges in doing so with AI. We took steps to mitigate biases in the models by testing multiple models and prompts to avoid singular biased outputs and avoided inferring protected attributes. We did not deploy the models in any public-facing system; all experiments were run offline with human oversight and conservative thresholds. The risk of false positives/negatives in misinformation detection is discussed in the paper, and we stress that any AI-generated labels should be vetted by human experts before any enforcement action. To balance reproducibility with privacy, we release evaluation assets (prompts, code, judge templates, redacted logs) while access to raw Weibo posts follows the prior study's protocol, and we document compute settings at a high level.

## Broader Impact Statement

The broader impacts of this work are potentially far-reaching in the ongoing fight against misinformation. On the positive side, our system could enable faster responses to false narratives during health crises, potentially saving lives by getting correct information to the public more quickly. It lowers the barrier for resource-limited health authorities to monitor and respond to social media trends.

However, negative impacts must be mitigated: heavy reliance on AI judgment can raise issues of censorship, surveillance, and free expression, especially if the models have hidden biases or error patterns. To mitigate this, we envision our tool as assisting human fact-checkers, not replacing them, with documented auditing and appeal procedures. We make our evaluation assets public (prompts, code, judge templates, redacted logs) to support scrutiny; access to raw Weibo posts follows the prior study's protocol, and we discourage uses that target individuals or predict protected attributes. We also report resource usage at a coarse level and favor efficient settings.

## Reproducibility Statement

We facilitate reproducibility while respecting data governance:

1. We make our evaluation assets public (prompts, code, judge templates, redacted logs); access to raw posts follows the prior study's protocol.

2. Our generation and judging code is provided with instructions to replicate API calls.

3. We report key hyperparameters and runtime model identifiers in the main paper.

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

# A    Technical Appendix

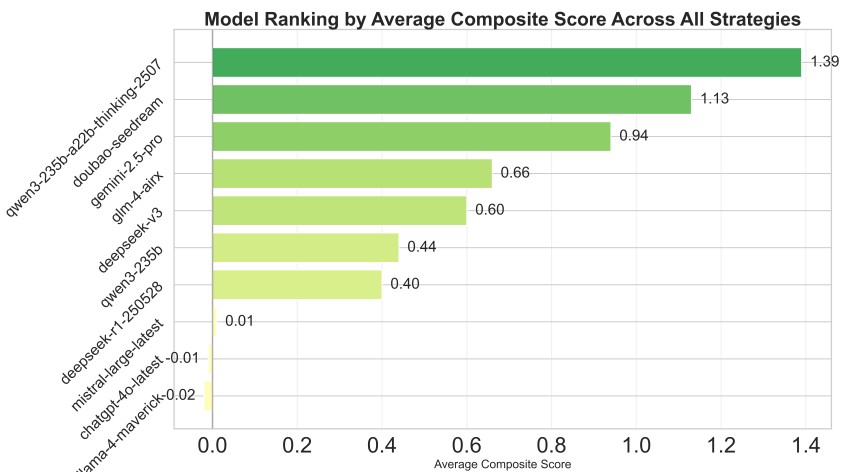

Figure A1: Model ranking by average composite score across strategies. Bars rank models by the mean composite score (higher is better; ambiguous = 0, correct rejections weighted more than mild judgments). The zero line indicates break-even (correct vs. incorrect judgments balanced).

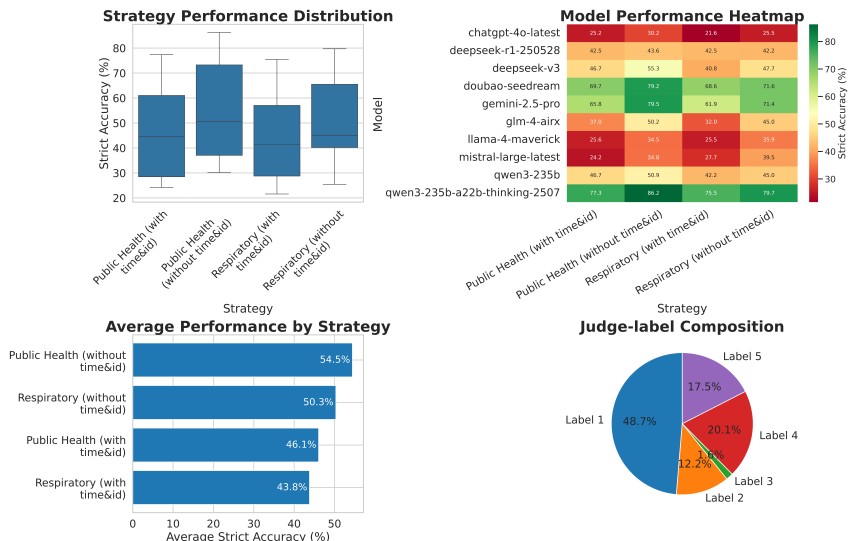

Figure A2: Composite comparison figure showing heatmap, box plot, bar chart, and pie chart of results.


