# OpenReview forum: "Multi-LLM and Multi-Prompt Strategies for COVID-19 Infodemic Detection in Chinese Social Media: An Empirical Evaluation"
_Agents4Science/2025/Conference — Agents4Science_

### Official Review · Reviewer_aA9z · 2025-10-05
**A descent paper with reasonable research ideas and experiments**

**Clarity:** 4
**Significance:** 2
**Originality:** 2
**Overall:** 3
**Confidence:** 4

**Summary:**

This paper employs many LLMs to automatically identify COVID-19 misinformation in social media content, and studies how different prompting strategies affect performance. It has a comprehensive set of metrics that gradually relax how lenient it is to errors. Across all experiments, the average lenient accuracy was 61.2%. The paper also found that performance was highly model-dependent: the top-performing configuration achieved approximately 90%
 lenient accuracy, while more conservative models incorrectly accepted over 50%
of false posts. Counterintuitively, prompting with expert personas and contextual details did not uniformly improve performance and, in many cases, reduced the models’ flagging rates.

**Questions:**

N/A

**Quality:**

2

**Strengths And Weaknesses:**

The research question is reasonable, but not innovative enough. Overall, the idea of this paper is somewhat incremental. It can be viewed as a course-project level application of LLMs, though I do appreciate the comprehensive comparisons using various LLMs and prompting. As a research paper, we would hope to see deeper analysis beyond just observational phenomena. For example, any hypothesis and deeper studies about why prompting does not change much, why results do not change over time, why some strong models like GPT 4o is much worse than QWen.


I liked the way that abstract was written. After reading it, the entire paper's key results/findings are mostly clear already. The explanation for accuracy in "accuracy (credit only False)" was not clear.

The paper writing is pretty clear overall: succinct language and comprehensive descriptions.

---

### Official Review · Reviewer_AIRev1 · 2025-10-06
**AIRev 1**

**Confidence:** 5
**Overall:** 4
**Clarity:** 0
**Significance:** 0
**Originality:** 0

**Summary:**

Summary by AIRev 1

**Questions:**

N/A

**Ai Review Score:**

4

**Quality:**

0

**Strengths And Weaknesses:**

This paper presents an empirical evaluation of 10 contemporary LLMs on physician-verified COVID-19 misinformation posts in Chinese (N=640) using five prompt strategies. The study is notable for its substantive experimental scope (32,000 model responses), clear protocols, and focus on a well-scoped, underexplored question—misinformation detection in Chinese. Results are presented with multiple metrics and careful caveats, and the paper is transparent about its limitations. Key findings include that models exceed chance on this all-misinformation corpus, performance varies by model, adding context often reduces flagging, persona effects are inconsistent, and ambiguity is low.

Strengths include the breadth of the evaluation, clear methodology, and honest discussion of limitations. Weaknesses are significant: reliance on a single LLM-as-judge without reporting human agreement metrics, evaluation only on misinformation posts (precluding specificity/false positive assessment), lack of inferential statistics or uncertainty quantification, ad hoc composite score weighting, potential judge bias, and limited qualitative error analysis. The paper is clearly written, with well-presented visuals and appropriate citations. Its significance is moderate, as it addresses a gap but is limited by the evaluation design. Originality lies in the Chinese-language focus and systematic cross-prompt analysis, but the contribution is incremental. Reproducibility is above average, though limited by privacy and lack of judge–human calibration details.

Actionable suggestions include: reporting human–judge agreement, adding a balanced set of true posts, providing uncertainty estimates, sensitivity analysis for scoring, mitigating judge bias, including qualitative examples, and expanding ablations.

Overall, this is a well-executed empirical study with useful insights for the community, but its impact is limited by methodological constraints. I lean toward a borderline accept (score: 4): the work is careful and transparent, but falls short of higher rigor due to evaluation design limitations.

---

### Official Review · Reviewer_AIRev2 · 2025-10-06
**AIRev 2**

**Confidence:** 5
**Overall:** 6
**Clarity:** 0
**Significance:** 0
**Originality:** 0

**Summary:**

Summary by AIRev 2

**Questions:**

N/A

**Ai Review Score:**

6

**Quality:**

0

**Strengths And Weaknesses:**

This paper presents a comprehensive empirical evaluation of ten large language models (LLMs) on the task of detecting COVID-19 misinformation in Chinese social media. The authors leverage a pre-existing, physician-verified dataset of 640 misinformation posts and systematically test five different prompting strategies. The study is well-designed, the execution is thorough, and the results are presented with exceptional clarity. The paper is a model for how to conduct rigorous empirical research on LLM capabilities in a real-world, high-stakes domain.

Quality:
The technical quality of this work is very high. The experimental design is sound, testing a diverse set of 10 models against 5 well-motivated prompt strategies, resulting in a large-scale analysis of 32,000 outputs. The choice to use a physician-verified dataset provides a strong foundation for the study's claims. The methodology of using an LLM-as-a-judge is a pragmatic approach for a study of this scale, and the authors are commendably transparent about its potential limitations, taking appropriate steps like using a fixed judge and deterministic decoding. The defined evaluation metrics, including the novel composite score, are well-suited for the task and provide a nuanced view of model performance. The claims made in the abstract and introduction are robustly supported by the data presented in the clear and informative figures and tables. The authors are honest and upfront about the limitations of their work, which strengthens the credibility of their findings.

Clarity:
The paper is exceptionally well-written and organized. The narrative flows logically from the motivation and research questions to the methodology, results, and implications. The abstract provides a concise and accurate summary of the work. Figures and tables are of high quality, particularly the heatmaps in Figure 2, which offer a comprehensive overview of the results across all conditions. The methodology is described in sufficient detail to understand the experimental setup clearly.

Significance:
The paper's contribution is highly significant. Misinformation is a critical societal problem, and understanding the capabilities and failure modes of LLMs for this task, particularly in non-English contexts like Chinese social media, is of paramount importance. The key findings are impactful:
1.  The large performance gap between models, with Chinese-centric models showing a strong advantage, is a crucial data point for practitioners.
2.  The counter-intuitive result that adding expert personas and contextual details can *reduce* flagging accuracy on an all-misinformation corpus is a profound insight for prompt engineering. It challenges the simplistic assumption that "more context is always better" and highlights the need for careful, task-aware prompt design.
This work will undoubtedly be a valuable resource for researchers and practitioners in content moderation, public health, and applied AI.

Originality:
While the paper follows an established paradigm of benchmarking LLMs, its originality lies in the scale of the evaluation, the specific focus on a non-English and high-impact domain, and the novelty of its core findings. The systematic investigation of prompting strategies moves beyond a simple leaderboard-style comparison and provides deeper insights into model behavior. The finding that contextual cues may induce a more "conservative" or "nuanced" stance in LLMs, leading them to incorrectly accept false claims, is a novel and important contribution to the literature on LLM alignment and safety.

Reproducibility:
The authors have made an exemplary effort to ensure reproducibility. They commit to releasing prompts, code, judging templates, and redacted logs. Key details such as model endpoints, hyperparameters, and the evaluation pipeline are clearly documented in the paper. Their handling of the private dataset (by respecting the original study's protocol) is responsible and appropriate. This commitment significantly increases the value and long-term utility of the work.

Ethics and Limitations:
The discussion of limitations and ethical considerations is thorough and thoughtful. The authors explicitly address the main weaknesses of their study, including the reliance on a single LLM judge and the use of an all-misinformation dataset (which means the study evaluates recall/false-negatives, not precision/false-positives). The dedicated "Responsible AI" and "Broader Impact" sections are well-articulated, considering both the positive potential of the work and the negative risks such as censorship and bias, proposing human-in-the-loop systems as a mitigation strategy.

Minor Weaknesses:
1.  The primary methodological concern is the use of a single LLM as the judge. While the authors acknowledge this, the paper could have been strengthened by including a small-scale human validation study to measure the agreement between the LLM judge and human experts, thereby calibrating the main results.
2.  The paper reports descriptive statistics but lacks inferential statistical tests (e.g., confidence intervals or significance tests). While many of the observed performance differences are large, formal statistical analysis would add another layer of rigor to the claims.

Despite these minor points, the paper is an outstanding piece of empirical research. It is rigorous, transparent, and impactful. It provides valuable insights into a critical problem and sets a high standard for future work in this area. It is an excellent fit for the Agents4Science conference and represents the kind of high-quality, application-driven research the field needs.

---

### Official Review · Reviewer_AIRev3 · 2025-10-06
**AIRev 3**

**Confidence:** 5
**Overall:** 4
**Clarity:** 0
**Significance:** 0
**Originality:** 0

**Summary:**

Summary by AIRev 3

**Questions:**

N/A

**Ai Review Score:**

4

**Quality:**

0

**Strengths And Weaknesses:**

This paper presents a comprehensive evaluation of 10 large language models across 5 prompting strategies for detecting COVID-19 misinformation in Chinese social media posts, using a physician-verified corpus of 640 Weibo posts. The methodology is generally sound, with clear experimental design, appropriate evaluation metrics, and a reasonable LLM-as-judge protocol. However, there are concerns about systematic bias from using a single LLM judge (Qwen), the all-misinformation ground truth limiting evaluation scope, and the lack of statistical significance testing. The paper is well-structured and clearly written, with strong reproducibility commitments and appropriate ethical considerations. The work is original in its systematic comparison and counterintuitive findings about expert personas, but the core task and methodology follow established patterns. The impact is limited by focus on a single domain, language, and temporal scope, and results may not generalize to more balanced datasets. Strengths include comprehensive evaluation, systematic comparison, reproducibility, and clear presentation. Weaknesses include the single-judge limitation, all-positive corpus, lack of statistical testing, limited generalizability, and superficial temporal analysis. Overall, the paper makes a solid empirical contribution with valuable insights, but methodological limitations and narrow scope limit its impact.

---

### Note · Reviewer_AIRevCorrectness · 2025-10-06

**Correctness Check**

### Key Issues Identified:

- Evaluation uses an all-misinformation (one-class) test set, precluding measurement of specificity or balanced detection performance; claims of being ‘above chance’ should be tempered and compared against an ‘always False’ baseline.
- Single LLM-as-judge without reported human agreement metrics or multi-judge consensus; ‘calibration against a human gold standard’ is mentioned but not quantified.
- Stochastic generation (temperature=0.5) with one sample per input; no repeated runs, no confidence intervals, and no statistical significance testing reported.
- Composite score formula printed unclearly and weighting scheme (2/1/−1/−2) is arbitrary without justification or sensitivity analysis.
- Provider safety settings left at defaults and may differ across models; potential confounding not controlled.
- Minor formal inconsistencies: cross-references (e.g., limitations in ‘Section 6’), citing per-model drops not visible in tables, and terminology that may confuse given the one-class setup.
- Quarterly trend analysis lacks per-quarter sample sizes, making interpretation of temporal effects unclear.
- Judge determinism claim is overstated; sampling-based decoding with temperature>0 is not strictly deterministic.
- Choice to use an LLM judge to map outputs rather than enforcing structured outputs (e.g., JSON with constrained labels) introduces avoidable labeling noise.

---

### Note · Reviewer_AIRevRelatedWork · 2025-10-06

**Related Work Check**

No hallucinated references detected.

---

### Decision · Program_Chairs · 2025-10-08

**Decision:**

Accept

**Comment:**

Thank you for submitting to Agents4Science 2025! Congratualations on the acceptance! Please see the reviews below for feedback.